# Evaluation of Chemical Compositions and the Antioxidant and Cytotoxic Properties of the Aqueous Extract of Tri-Yannarose Recipe (*Areca catechu*, *Azadirachta indica*, and *Tinospora crispa*)

**DOI:** 10.3390/antiox12071428

**Published:** 2023-07-15

**Authors:** Sineenart Sanpinit, Palika Wetchakul, Piriya Chonsut, Nuntika Prommee, Chuchard Punsawad, Jaehong Han, Soiphet Net-anong

**Affiliations:** 1Department of Applied Thai Traditional Medicine, School of Medicine, Walailak University, Nakhon Si Thammarat 80160, Thailand; 2Research Center in Tropical Pathobiology, Walailak University, Nakhon Si Thammarat 80160, Thailand; 3Department of Applied Thai Traditional Medicine, Faculty of Public Health, Naresuan University, Phitsanulok 65000, Thailand; 4Department of Medical Sciences, School of Medicine, Walailak University, Nakhon Si Thammarat 80160, Thailand; 5Metalloenzyme Research Group, Department of Plant Science and Technology, Chung-Ang University, 4726 Seodong-daero, Anseong 17546, Republic of Korea

**Keywords:** Thai herbal formula, Thai traditional medicine, LC-QTOF-MS, water extract, chemical constituents, oxidative stress

## Abstract

Tri-Yannarose is a Thai traditional herbal medicine formula composed of *Areca catechu*, *Azadirachta indica*, and *Tinospora crispa*. It possesses antipyretic, diuretic, expectorant, and appetite-stimulating effects. This study aimed to evaluate the antioxidant activities, cytotoxicity, and chemical constituents of an aqueous extract following a Tri-Yannarose recipe and its plant ingredients. The phytochemical analysis was performed using LC-QTOF-MS. Antioxidant activities were determined using DPPH, ABTS, TPC, TFC, FRAP, NBT, MCA, and ORAC assays. Cytotoxicity was investigated using a methyl thiazol tetrazolium (MTT) assay. In addition, the relationship between the chemical composition of Tri-Yannarose and antioxidant activities was investigated by examining the structure–activity relationship (SAR). The results of the LC-QTOF-MS analysis revealed trigonelline, succinic acid, citric acid, and other chemical constituents. The aqueous extract of the recipe showed significant scavenging effects against ABTS and DPPH radicals, with IC_50_ values of 1054.843 ± 151.330 and 747.210 ± 44.173 µg/mL, respectively. The TPC of the recipe was 92.685 mg of gallic acid equivalent/g of extract and the TFC was 14.160 mg of catechin equivalent/g of extract. All extracts demonstrated lower toxicity in the Vero cell line according to the MTT assay. In addition, the SAR analysis indicated that prenyl arabinosyl-(1–6)-glucoside and quinic acid were the primary antioxidant compounds in the Tri-Yannarose extract. In conclusion, this study demonstrates that Tri-Yannarose and its plant ingredients have potent antioxidant activities with low toxicity. These results support the application of the Tri-Yannarose recipe for the management of a range of disorders related to oxidative stress.

## 1. Introduction

From the past till now, Thai traditional medicine (TTM) has been used in daily life in Thailand for disease treatment and health, both as single herbs and as recipes. Tri-Yannarose (TYN) is a recipe from Thai traditional medicine that is composed of the core or inner trunk of *Areca catechu* L. (AC), the root of *Azadirachta indica* A. Juss. var. siamensis Valeton. (AI), and the vines of *Tinospora crispa* (L.) Hook. f. & Thomson (TC). This recipe has been employed as a treatment to alleviate fever and promote urination, as an expectorant and appetite stimulant, and as a medicine to nourish the fire element [1]. A previous study discovered that Tri-Jannarose, a similar recipe to Tri-Yannarose, exhibited antioxidant activity. Tri-Jannarose consists of the seeds of *A. catechu*, the root of *A. indica*, and the vines of *T. crispa*. That study showed that Tri-Jannarose contained phenolic and flavonoid compounds, which are strong potent antioxidants, and had a high α-glucosidase inhibitory activity [2]. In this study, different parts of the same plants were used, i.e., the inner trunk instead of the seeds of *A. catechu*, and *T. crispa* vines without leaves. However, the part of AC used in this recipe is still controversial and is not clear. Some Thai folk medicine healers use the seeds of AC, but other healers use the inner trunk of AC. For this study, we used the inner trunk of AC in the preparation of the TYN recipe. 

A literature review revealed several active constituents of AC, including alkaloids, polyphenols, tannins, flavones, triterpenes, steroids, fatty acids, minerals, and phenolic compounds [3,4,5,6]. Furthermore, several parts of AC, including leaves, fruits, and seeds, have a wide range of pharmacological activities, including antioxidative (preventing oxidative damage in normal cells) [7,8], anti-inflammatory [7], antihyperglycemic [9], and antiparasitic activities [6]. However, the biological activity of the inner trunk of AC has rarely been reported. A previous study [10] reported that the root and bark of AI had a high antioxidant activity, and another study showed that the ethanolic extract of dried TC stems had a strong antioxidant activity [11]. AI contains a variety of components, such as azadirachtin, nimbin, nimbidin, nimbolide, quercetin, and limonoids. Diterpenoids from the roots of AI have been shown to have antibacterial, antifungal, and anti-inflammatory effects [12]. Earlier studies have confirmed their roles as antioxidants [13], inhibitors of angiogenesis [14], and antidiabetics [15]. Decoction of the stems of TC has been employed as an antipyretic in Thai folk treatments to reduce thirst, boost appetite, decrease body temperature, and preserve good health [16,17]. The chemical constituents of TC contain more than 65 compounds, including lactones, steroids, flavonoids, lignans, and furanoditerpenes [18]. Several scientific studies have found that it has antioxidant [19,20], anti-inflammatory [21], antimalarial [22], and immunomodulatory effects [23]. 

Antioxidants are compounds in foods or herbs that may prevent or delay oxidation, which causes free radicals. Under certain conditions, the number of free radicals is too high for the antioxidant system to handle. This causes a condition known as “oxidative stress”, which affects cells in the body, resulting in aging and the development of various chronic diseases, such as coronary artery disease, cardiovascular disease, immune-related diseases, and cancer [24]. The body has two mechanisms for getting rid of free radicals: utilizing enzymes in the body, such as superoxide dismutase (SOD), and without using enzymes, such as vitamin E (alpha-tocopherol), beta-carotene, and vitamin C. It has been suggested that the long-term consumption of foods rich in antioxidants can delay or prevent the conditions of oxidative stress [25]. 

However, there is less evidence to support the use of the Tri-Yannarose recipe in medicine and healthcare. Therefore, the purpose of this study was to evaluate the antioxidant and cytotoxic characteristics of the extracts of TYN and its plant ingredients, as well as to analyze their chemical contents using LC-QTOF-MS to confirm their pharmaceutical properties. In addition, the relationship between the chemical composition of Tri-Yannarose and its antioxidant activities was investigated by examining the structure–activity relationship (SAR). 

## 2. Materials and Methods

### 2.1. Chemicals

Dihydrochloride (AAPH), 2,2,-Azinobis (3-ethylbenzothiazoline-6-sulfonic acid) (ABTS), 2,2′-azobis-2-methyl-propanimidamide, 2,2-Diphenyl-1-picrylhydrazyl (DPPH), 2,4,6-Tris(2-pyridyl)-s-triazine (TPTZ), catechin hydrate, nitrotetrazolium blue chloride, phosphate-buffered saline (PBS), and Trolox were acquired from Sigma-Aldrich (St. Louis, MO, USA). Acetic acid and dimethyl sulfoxide (DMSO) were purchased from RCI Labscan (Bangkok, Thailand). Sodium carbonate (Na_2_CO_3_), aluminum chloride (AlCl_3_), sodium hydroxide (NaOH), potassium persulfate (K_2_S_2_O_8_), ethylenediaminetetraacetic acid (EDTA), iron (III) chloride hexahydrate (ferric chloride), and sodium nitrite (NaNO_2_) were purchased from KAMAUS (New South Wales, Australia). Gallic acid was purchased from Acros Organics (Geel, Belgium). Solvents, including methanol, ethyl acetate, and ethanol, were purchased from J. T. Baker (Radnor, PA, USA). Other agents utilized were analytical-grade substances that were bought from commercial sources. During sample preparation, dilution, and pre-analytical rinses, deionized water was utilized.

### 2.2. Plant Extract Preparation

The inner trunk of AC was harvested from a betel tree in Tha Sala, Nakhon Si Thammarat, Thailand. The root of AI and the vines of TC were purchased from a herbal pharmacy (Triburi Orsot) located in Hat Yai, Songkhla, Thailand. The plant samples were verified and placed at the Thai traditional medicine herbarium of the Department of Thai Traditional and Alternative Medicine, Ministry of Public Health, Thailand. The voucher specimen of AC is TTM-c No. 1000740; for AI, it is TTM-c No. 1000741, and for TC, it is TTM-c No. 1000742. After being thoroughly washed with running water, the fresh materials were dried in a hot air oven for 48 h at 60 °C before being ground into a fine powder using an electric grinder and sifted through a no. 80 sieve. The extract based on the TYN recipe was prepared by mixing AC, AI, and TC in a 1:1:1 ratio and boiled for 15 min in distilled water (1 kg:1 L). Additionally, the extracts of its plant ingredients, including AC, AI, and TC, were prepared by boiling 1000 g of each herb with 1 L of distilled water for 15 min. Following this, the samples were filtered and concentrated via freeze drying to obtain an aqueous extract. The aqueous extracts of all samples (TYN recipe, AC, AI, and TC) were dissolved with distilled water into various concentrations for analysis of their antioxidant activity. The following equation was used to compute the extraction yield of the TYN recipe, AC, AI, and TC: Extraction yield (%) = (weight of dry extracts/weight of the dry plant) × 100(1)

### 2.3. Analysis of the Chemical Elements of the TYN Recipe Using Liquid Chromatography–Quadrupole Time-of-Flight Mass Spectrometry (LC-QTOF-MS)

Using UHPLC, a Zorbax eclipse plus C18 quick resolution HD column (150 mm in length, 2.1 mm in inner diameter, and particle size of 1.8 m) was employed to investigate the chemical composition of the TYN recipe’s aqueous extract (6500 series Q-TOF system, Agilent technologies, Santa Clara, CA, USA). A 2 µL solution was injected into the column. The temperature was maintained at 35 °C. Elution was performed within 30 min. The mobile phase program consisted of 0.1% formic acid in water (A) and acetonitrile (B) at a flow rate of 0.2 mL/min. The gradient was initiated with solvent A: solvent B (90%:10%) for 0–15 min, then automatically changed to solvent A: solvent B (0%:100%) for 15–20 min, and finally automatically changed to solvent A: solvent B (90%:10%) for 20–30 min. A dual AJS ESI ion source with a scanning range of *m*/*z* 50 to 1500 was utilized to conduct reference masses in both positive and negative ESI modes with the following conditions: a capillary voltage of 4000 V, a nozzle voltage of 2000 V, and a gas flow rate of 13 L/min at 325 °C, and nebulization was set as 35 psi while 10, 20, and 40 eV collision energies were used. The workflows were qualitatively analyzed using the MassHunter workstation software Version B.08.00.

### 2.4. Quantification of Phytochemical Contents

#### 2.4.1. Total Phenolic Content

The total phenolic content was determined using the Folin–Ciocalteu technique with a minor modification, as previously described [26]. Briefly, 120 µL of each sample at a concentration of 2.5 mg/mL was mixed in the dark for 5 min with 1 mL of Folin–Ciocalteu dilution before being neutralized with 125 µL of 20% *w*/*v* Na_2_CO_3_ solution. The mixture was incubated in the dark for 90 min at room temperature. A microplate reader was used to measure absorbance at 725 nm. The results were calculated using the gallic acid calibration curve and presented as mg of gallic acid equivalent/g of extract (mg of GAE/g of extract).

#### 2.4.2. Total Flavonoid Content

The total flavonoid content was determined using the aluminum chloride hexahydrate colorimetric technique [27]. Briefly, 20 µL of each sample with a concentration of 2.5 mg/mL and 6 µL of 5% *w*/*v* NaNO_2_ were mixed. After 6 min, 6 µL of 10% *w*/*v* AlCl_3_ was added. Following this, 40 µL of 4% *w*/*v* NaOH and 48 µL of distilled water were added and mixed for 6 min. A microplate reader was used to measure absorbance at 510 nm. The results were reported as mg of catechin equivalent/g of extract (mg of CE/g of extract) using the catechin calibration curve. 

### 2.5. Chain-Breaking Antioxidant Capacity

The chain-breaking antioxidant activity of the samples was assessed utilizing a range of methodologies with various modes of action.

#### 2.5.1. Mixed-Mode-Based Assays

The 2,2-Diphenyl-1-picrylhydrazyl (DPPH) radical scavenging assay was applied following Brand-Williams et al., 1995 [28]. In a microplate, each sample solution (20 µL) at a concentration ranging from 1.22 µg/mL to 2500 µg/mL and the DPPH working solution (180 µL) were mixed. The plate was incubated in the dark for 30 min at room temperature, and the absorbance of the solution at 490 nm was measured using a microplate reader.

The 2,2,-Azinobis (3-ethylbenzothiazoline-6-sulfonic acid) (ABTS) radical scavenging assay was applied following Wetchakul et al., 2022 [27]. Firstly, an ABTS radical cation (ABTS^•+^) was created by combining 2 mM of ABTS and 2.45 mM of K_2_S_2_O_8_ in a volume ratio of 1:1. The mixture was allowed to stand in the dark for 16 h at room temperature. ABTS^•+^ must be diluted with ethanol before use to obtain an absorbance of 0.70 ± 0.05 at 734 nm. Following this, in a 96-well plate, 20 µL of each sample and 180 µL of the ABTS^•+^ solution were added. After the initial mixing, the mixture was incubated at room temperature for 6 min to measure absorbance at 734 nm.

The DPPH or ABTS radical scavenging activity was reported when a 50% inhibitory concentration (IC_50_; mg/mL) of DPPH or ABTS radicals was obtained. Trolox was utilized as the standard antioxidant chemical.

#### 2.5.2. Single-Electron-Transfer-Mechanism-Based Assays

According to a previous work [29], the ferric reducing antioxidant power (FRAP) activity of the plant extracts was determined by converting ferric (Fe^3+^) to ferrous (Fe^2+^). Each well of a 96-well microtiter plate was filled with 20 µL of each sample (0.078 mg/mL), mixed with 180 µL of the FRAP reagent, and incubated in the dark for 30 min at room temperature. The colored constituent of an intense blue color complex generated at 593 nm was used to characterize the solution’s absorbance. For the standard curve, a freshly prepared working solution of FeSO_4_ was used. The result is presented as µM of Fe_2_SO_4_ per milligram of extract.

The riboflavin/methionine/illuminate system produces superoxide anion radicals, which are then measured using the nitroblue tetrazolium (NBT) assay [30]. The generated NBT is converted to purple formazan (NBT^2+^). A total of 30 μL of each sample or catechin (positive control) was added to a 96-well microtiter plate. Next, 30 μL of riboflavin (30 μg/mL), methionine (30 μg/mL), and EDTA (20 μg/mL) were added. Finally, NBT (400 μg/mL) at 30 μL was added. Fluorescent lamps (20 W) were lighted for 25 min at 25 °C to take the step of photo-induced superoxide radicals. Absorbance was measured at 560 nm after incubation. The superoxide radical scavenging activity was represented as the 50% inhibitory concentration (IC_50_; mg/mL).

#### 2.5.3. Hydrogen Atom Transfer (HAT)-Based Assay

Peroxyl radical scavenging activity was measured using the ORAC test. Thermal homolysis of 2,2′-azo-bis-(2-amidinopropane) dihydrochloride (AAPH) generates peroxyl radicals [31]. This test was carried out on 96-well plates with black walls using phosphate-buffered saline (PBS) at a pH of 7.4. Each sample was tested at 25 µL with a previous concentration ranging from 2.0 to 100 µg/mL, and the benchmark was 25 µL of Trolox solution (2-fold dilution). All sample wells received 150 µL of sodium fluorescein at a concentration of 40 mM. After incubation for 3 min at 30 °C, 25 µL of the AAPH solution was added to the mixture. Fluorescence was measured every five minutes for 90 min using a microplate reader with excitation at 485 nm and emission at 528 nm. The amount of Trolox equivalents per gram of extract (M of TE/g of extract) is the unit used to express the antioxidant capacity. Each sample’s area under the curve (AUC) was computed, along with a comparison to Trolox. The AUC equations are as follows:AUC = (R_1_/R_1_) + (R_2_/R_1_) + (R_3_/R_1_) + … + (R_4_/R_1_)(2)
NET AUC = AUC _sample_ − AUC _blank_(3)

In the above equation, R1 represents the initial fluorescence reading and Rn represents the end measurement.

### 2.6. Metal-Chelating Activity

The ability of the polyherbal extracts to chelate ferrous ions was evaluated using an established colorimetric metal-chelating-activity (MCA) technique [32]. As a positive control, ethylenediaminetetraacetic acid (EDTA) complex was utilized. In this test, 0.4 mL of 0.25 mM ferrozine was used to initiate the reaction before 0.2 mL of the plant extracts, with a concentration ranging from 0.03 to 62.50 mg/mL, was added. After standing at room temperature for 10 min, the stable ferrous–ferrozine combination showed an improvement in absorbance at 562 nm. MCA was presented as the concentration that provided 50% inhibition (IC_50_) of the ferrous ion–ferrozine complex and was computed using the following equation:Metal chelating activity (%) = (Abs _control_ − Abs _test_/Abs _control_) × 100(4)

The IC_50_ value, indicating the concentration at which 50% chelating activity is inhibited, demonstrates a higher metal-chelating activity as its value decreases.

### 2.7. 3-[4,5-Dimethylthiazol-2-yl]-2,5 Diphenyl Tetrazolium Bromide (MTT) Assay

Vero cells were seeded at a density of 1 × 10^5^ cells/well in 96-well plates. The plates were cultured in an incubator at 37 °C overnight before being treated for 24 h. Each sample was compared to doxorubicin (Dox) as a positive control at doses of 5, 10, 20, 40, and 80 µg/mL [33]. MTT was added to the wells and then incubated for 4 h at 37 °C. Via the use of a microplate reader, the absorbance of each well was measured at 540 nm. The percentage of cell viability was estimated using the following equation:% Cell viability = [(Mean OD of treated wells)/(Mean OD of untreated wells)] × 100(5)

### 2.8. Structure–Activity Relationship (SAR)

Predictions of biological activity for the Tri-Yannarose recipe were performed using the PASS server, as described in previous studies [34,35]. This test was performed to assess oxidant inhibitory activity. Based on the SAR analysis, the program classifies a substance’s anticipated activity spectrum as either “probably active” (Pa) or “probably inactive” (Pi). The following are some of the ways in which the PASS prediction findings could be interpreted: A molecule’s viability is determined according to whether its Pa value is larger than its Pi value. If the value of Pa is greater than 0.70, then it is likely that the activity may be detected experimentally. If Pa is less than 0.5, the likelihood of discovering a structurally novel chemical component increases, while the likelihood of discovering an activity experimentally decreases. If Pa is greater than 0.7, the likelihood of discovering an activity experimentally increases while the similarity to known pharmacological drugs decreases.

### 2.9. Statistical Analysis

Each test was carried out in triplicate. The data were analyzed using SPSS (version 23) and represented as mean and standard deviation (SD). One-way ANOVA was used to determine significant differences between the samples, followed by Dunnett’s multiple comparison test, or Duncan’s test. The significance level was set at *p* < 0.05.

## 3. Results

### 3.1. Plant Extract Yield

The aqueous extract of Tri-Yannarose yielded 3.35 g/100 g of dry plant components. *T. crispa* showed the highest yield at 9.97%, followed by *A. catechu* (5.89%) and *A. indica* (0.57%).

### 3.2. Chemical Constituents as Determined by LC-QTOF-MS

The untargeted screening and identification of chemical components in the aqueous extract of Tri-Yannarose were carried out using LC-QTOF-MS in both positive and negative modes. Using the fragmentation patterns observed in LC-QTOF-MS in the positive mode, eight chemical components were identified (Table 1), including guanine, trigonelline, phytosphingosine, and trimeprazine. Fourteen chemical constituents were identified using the fragmentation patterns observed in LC-QTOF-MS in the negative mode (Table 2), including succinic acid, citric acid, epicatechin, and quinic acid. The LC-QTOF-MS chromatograms of the Tri-Yannarose recipe’s aqueous extract in the positive and negative modes are shown in Figure 1 and Figure 2, respectively. Moreover, The mass spectra of compound identification of Tri-Yannarose recipe’s aqueous extract identified via LC-QTOF-MS in the positive and negative mode based on the match scores obtained during the library search shown in Appendix A.

### 3.3. Total Phenolic and Total Flavonoid Contents of Tri-Yannarose and Its Plant Ingredients

The *A. indica* extract had the highest level of phenolics (320.146 mg of gallic acid equivalent/g of extract), while the *A. catechu* extract had the lowest (92.084 mg of gallic acid equivalent/g of extract), with the values of the Tri-Yannarose extract (92.685 mg of gallic acid equivalent/g of extract), and the *T. crispa* extract (94.360 mg of gallic acid equivalent/g of extract) being in between (Table 3). The flavonoid content ranged from 10.889 to 76.375 mg of catechin equivalent/g of extract (Table 3).

### 3.4. Antioxidant Activity of the Tri-Yannarose Recipe and Its Plant Ingredients as Determined by Mixed-Mode-Based Assays

The DPPH technique is one of the most widely used antioxidant assays because it is a rapid, reliable, and reproducible method for assessing the general antioxidant activity of natural compounds as well as plant extracts in vitro. According to the research of Marjoni and Zulfisa, 2003, the power levels of antioxidant activity can be divided into five levels: very high with an IC_50_ < 50 μg/mL, active with an IC_50_ of 50–100 μg/mL, moderately active with an IC_50_ of 101–250 μg/mL, less active with an IC_50_ of 250–500 μg/mL, and inactive with an IC_50_ > 500 μg/mL [36]. Our results indicated that the aqueous extract of Tri-Yannarose had inactive DPPH radical scavenging activity with an IC_50_ value of 747.210 ± 44.173 µg/mL. In addition, the aqueous extract of *A. indica* (highly active) exhibited stronger antioxidant activity than *A. catechu* (weak) and *T. crispa* (inactive), with an IC_50_ value of 49.461 ± 3.275 µg/mL (Table 4). The results of the ABTS assay showed that the IC_50_ value of the aqueous extract of Tri-Yannarose was 1054.843 ± 151.330 µg/mL. The aqueous extract of *A. indica* had the highest IC_50_ value (197.516 ± 6.036 µg/mL), followed by the *A. indica* (853.326 ± 160.301 µg/mL) and *T. crispa* (2288.524 ± 49.056 µg/mL) extracts, respectively. The aqueous extract of Tri-Yannarose and its constituents showed lower antioxidant activities than the standard according to the DPPH and ABTS assays (Table 4). 

### 3.5. Antioxidant Activity of the Tri-Yannarose Recipe and Its Plant Ingredients as Determined by Single-Electron-Transfer-Mechanism-Based Assays

Table 5 displays the findings of the nitroblue tetrazolium (NBT) dye reduction assay performed using the aqueous extracts of Tri-Yannarose and its plant ingredients. The results indicated that the aqueous extract of Tri-Yannarose had an IC_50_ value of 389.050 ± 3.110 µg/mL. The aqueous extract of *A. indica* showed no significant difference from the *A. catechu* extract. However, the aqueous extract of *T. crispa* showed a lower activity. The aqueous extracts of Tri-Yannarose and its plant ingredients showed a lower antioxidant activity than the standard based on the NBT assay.

Antioxidant activity evaluated using the FRAP technique was detected in all extracts (Table 5). The highest value was found in the *A. indica* extract, with 11,034.986 ± 207.094 mM of FeSO_4_/mg of extract, followed by the *A. catechu*, Tri-Yannarose, and *T. crispa* extracts, respectively. The quantity of total phenolics in the analyzed extracts, as determined by the Folin–Ciocalteau technique, ranged from 92.084 to 320.146 mg of gallic acid equivalent/g of extract (Table 5).

### 3.6. Antioxidant Activity of the Tri-Yannarose Recipe and Its Plant Ingredients as Determined by Hydrogen Atom Transfer (HAT)-Based Assay

The ORAC test is an important tool for determining the antioxidant capacity of extracts. The values of fluorescence-degradation inhibition demonstrated that the tested extracts scavenged peroxyl radicals in a concentration-dependent manner. The highest ORAC value was found for the *T. crispa* extract with 8.710 ± 0.559 μM of Trolox/gram of extract (Table 6). On the contrary, the aqueous extracts of Tri-Yannarose, *A. indica*, and *A. catechu* showed ORAC values of less than 1 μM of Trolox/gram of extract.

### 3.7. Metal Chelating Activity of the Tri-Yannarose Recipe and Its Plant Ingredients

The metal chelating activities (MCA) of the aqueous extracts of Tri-Yannarose and its plant ingredients were measured to assess their capacity to prevent metal–lipid interactions. The results showed that the aqueous extract of Tri-Yannarose (IC_50_ = 702.109 ± 73.005 µg/mL) and the *T. crispa* extract (IC_50_ = 670.110 ± 85.019 µg/mL) showed no significant differences in metal chelating ability. The aqueous extract of Tri-Yannarose and its constituents showed a lower metal chelating activity than the standard according to the MCA assay (Table 7). 

### 3.8. In Vitro Assessment of Cytotoxicity

The cytotoxicity of Tri-Yannarose, *A. catechu*, *A. indica*, and *T. crispa* aqueous extracts against the Vero cell line was evaluated, showing IC_50_ values of >100 g/mL, >300 g/mL, >500 g/mL, and >100 g/mL, respectively. All extracts were less hazardous than DOX (IC_50_ of 3.5 1.26 g/mL) in the Vero cell line, with IC_50_ values ranging from 100 to 1000 g/mL.

### 3.9. Structure–Activity Relationship (SAR) of the Tri-Yannarose Recipe

After conducting an analysis using the SAR testing criteria, the results revealed that compounds demonstrating a favorable activity should possess a Pa value greater than 0.7 and a Pi value as close to 0 as possible. Subsequently, an investigation into the structure–activity relationship of the Tri-Yannarose extract revealed the presence of two significant compounds exhibiting antioxidant activity: prenyl arabinosyl-(1–6)-glucoside with a Pa value of 0.842 and a Pi value of 0.003, and quinic acid with a Pa value of 0.830 and a Pi value of 0.003. Notably, no other compounds within the extract exhibited the appropriate structural characteristics for antioxidant activity. Hence, it is reasonable to hypothesize that prenyl arabinosyl-(1–6)-glucoside and quinic acid serve as the main antioxidant components in the Tri-Yannarose extract. Nevertheless, further investigations are required to validate the antioxidant efficacy of prenyl arabinosyl-(1–6)-glucoside and quinic acid derived from Tri-Yannarose extracts.

## 4. Discussion

The Thai traditional medicine, Tri-Yannarose, has a number of medicinal substances with three known ingredients: *A. catechu*, *A. indica*, and *T. crispa*. According to previous reports, *A. catechu* extracts have the capacity to prevent oxidative damage in normal cells due to their antioxidant properties [8]. One previous study [10] reported that the root and bark of *A. indica* had a high antioxidant activity, and another study showed that ethanolic extract from dried *T. crispa* stems had a strong antioxidant activity [11]. However, there was only one similar study that investigated antioxidant activity using the Tri-Jannarose recipe. This study utilized betel palm seeds (*A. catechu*), Siamese neem tree root (*A. indica*), and heart-leaved moonseed vines (*T. crispa*) as ingredients [2]. In our present study, the Tri-Yannarose recipe also incorporated *A. indica* root, but it used *T. crispa* vines without leaves and the inner trunk instead of the seeds of *A. catechu*; in Thai traditional medicine, it is not clearly explained which part of *A. catechu* is used to make the medicine. 

The TPC and DPPH values from earlier investigations were similar to those of this experiment (51.15 ± 0.421 mg of GE/gExt and 0.0002 ± 0.421 mg of GE/gExt, respectively), whereas the TFC values were lower (0.0002 ± 2.857 mg of QE/gExt) [2]. Different standard chemicals might have been used. Based on the outcomes of both investigations, antioxidant activity remained relatively constant. As a result, both betel palm seeds and the trunk of AC may be used in the Tri-Jannarose or Tri-Yannarose recipe to make medication.

This present study demonstrated that the aqueous extract of Tri-Yannarose showed significantly different scavenging effects against ABTS and DPPH radicals, with IC_50_ values of 747.210 ± 44.173 and 1054.843 ± 151.330 µg/mL, respectively (*p* < 0.05). This study is similar to a previous study that reported that the DPPH activity of the aqueous extraction of Tri-Jannarose was 1485 ± 5 µg/mL (inactive) [2]. Therefore, both Tri-Yannarose and Tri-Jannarose were inactive in their antioxidant activity when examined using the DPPH method [36]. This finding suggested that Tri-Yannarose includes molecules that are capable of donating hydrogen to a free radical to remove an odd electron, which is responsible for the radical’s reactivity [37]. The scavenging activity of the aqueous extract of Tri-Yannarose was greater for DPPH radicals than for ABTS radicals. Various parameters, such as (i) extract solubility in various testing systems, and (ii) functional groups contained in bioactive substances, have been shown to alter the capacity of extracts to react and quench different radicals [38]. Our findings showed that Tri-Yannarose was inactive scavenger for DPPH radicals and ABTS radicals. The total phenolic content of the aqueous extract of Tri-Yannarose was 92.685 mg of gallic acid equivalent/g of extract, and the total flavonoid content was 14.160 mg of catechin equivalent/g of extract. The other study using an aqueous extract of Tri-Jannarose presented a lower total phenolic content at 51.15 ± 0.421 mg of GE/gExt and a lower total flavonoid content at 0.0002 ± 2.857 mg of QE/gExt [2]. An analysis of Tri-Yannarose’s plant ingredients demonstrated that the AI extract had the highest total phenolic (*A. indica* > *T. crispa* > *A. catechu*) and total flavonoid contents (*A. indica* > *A. catechu* > *T. crispa*). It is possible that *A. indica* is the major herbal component that supports the antioxidant ability of the aqueous extract of Tri-Yannarose. The capacity of this Tri-Yannarose recipe to decrease the TPRZ-Fe (III) complex to TPTZ-Fe (II) was used to determine its antioxidant capability. Antioxidant activity is known to increase proportionally with polyphenol concentration. This action is thought to be mostly related to redox characteristics [39]. According to recent studies, total phenols and antioxidant activity appear to have a very favorable association in plant species [38]. This research showed that among the plant ingredients of Tri-Yannarose, the *A. indica* extract had the highest antioxidant activity (11,034.986 mM of FeSO_4_/mg of extract) according to the FRAP assay. *T. crispa* was reported to have a high antioxidant activity in previous studies [18,19,20]. In one study, the antioxidant activities of *A. indica* leaf and bark extracts were evaluated, and the results clearly demonstrated that all the tested neem leaf and bark extracts had considerable antioxidant activities [40]. 

This study discovered that all extracts tested for cytotoxicity in the Vero cell line were less dangerous than DOX. A previous study that determined the cytotoxicity of an aqueous extract of *A. catechu* nut via the MTT assay did not find cytotoxic activity in the Vero cell line according to the obtained IC_50_ values [41]. Similar to a previous report that determined toxicities in a normal cell line (MDBK), the ethanolic extract of *A. indica* seeds showed no toxic activity (IC_50_ > 650 μg/mL) [42]. A previous investigation found seven *A. indica* compounds with poor cytotoxicity against the normal lymphocyte cell line RPMI 1788 (IC_50_ = 16.1 M) when compared to cisplatin (IC_50_ = 1.9 M) [43]. In another study, the ethanolic extract from the stems of *T. crispa* had no cytotoxic effects on Vero cells, with an IC_50_ = 179.0 1.5 g/mL, when compared to clindamycin (IC_50_ = 116.5 2.3 g/mL) [44]. Furthermore, *T. crispa* was shown to have no cytotoxicity against Vero cells [45,46].

Our results showed that Tri-Yannarose and its herbal components have a good antioxidant capacity. Moreover, the chemical constituents in the aqueous extract of Tri-Yannarose were evaluated using the LC-QTOF-MS mass spectra. The results revealed 12 chemical constituents in the positive mode and 14 chemical constituents in the negative mode. One of the chemical constituents in the aqueous extract of Tri-Yannarose is trigonelline, a major alkaloid. Previous research has suggested that potential natural antioxidants might be employed in the treatment of illnesses, including diabetes mellitus [47]. In addition, based on the negative mode of LC-QTOF-MS, several organic acids were found in the aqueous extract of Tri-Yannarose, including succinic acid, citric acid, quinic acid, and isopalmitic acid. These organic acids can also play a biological role owing to their antioxidant, antimicrobial, and anti-inflammatory properties [48]. Furthermore, 3-dehydroshikimic acid (DHS), an intermediary in the biosynthesis of aromatic amino acids, was discovered in the aqueous extract of Tri-Yannarose. A previous study reported that DHS had a high antioxidant activity in lard, inhibiting peroxidation with an activity that was equivalent to tert-butylhydroquinone, propyl gallate, and gallic acid and superior to alpha-tocopherol [49]. From the previous reports, it is possible that some chemical constituents found in the aqueous extract of this Tri-Yannarose recipe support its antioxidant ability.

## 5. Conclusions

This study found that the aqueous extract of Tri-Yannarose and its plant ingredients have antioxidant activities, such as superoxide radical scavenging activity, hydroxyl radical scavenging activity, and activities to prevent oxidative stress in the body. The MTT test revealed that all plant ingredients used in the Tri-Yannarose recipe were less hazardous in the Vero cell line. The recipe contains a complicated chemical composition according to the LC-QTOF-MS fingerprint analysis, both in the negative and positive modes. Therefore, the evidence of this study supports the use of the Tri-Yannarose recipe for the treatment of a variety of diseases related to oxidative stress.

## Figures and Tables

**Figure 1 antioxidants-12-01428-f001:**
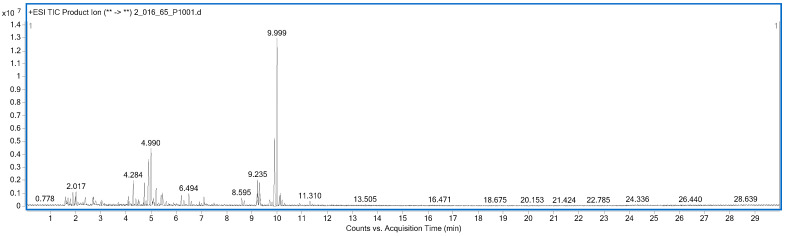
LC-QTOF-MS chromatogram of the Tri-Yannarose recipe’s aqueous extract in the positive mode based on the match scores obtained during the library search. (** -> **) refers to the disintegration of precursor ions into product ions.

**Figure 2 antioxidants-12-01428-f002:**
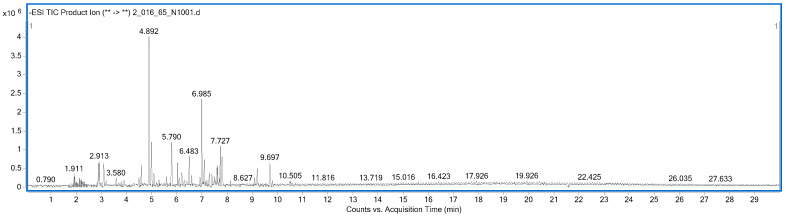
LC-QTOF-MS chromatogram of the Tri-Yannarose recipe’s aqueous extract in the negative mode based on the match scores obtained during the library search. (** -> **) refers to the disintegration of precursor ions into product ions.

**Table 1 antioxidants-12-01428-t001:** Proposed chemical components of the Tri-Yannarose recipe’s aqueous extract identified via LC-QTOF-MS in the positive mode based on the match scores obtained during the library search.

Compound Name	Formula	RT (min)	*m*/*z* Calculated	*m*/*z* Observed	Score (Lib)	Mass Error (ppm)	Fragmentation
Choline	C_5_H_14_NO	1.9	104.1079	104.1074	94.71	−3.82	((M)—C_5_H_14_NO)^+^
Trigonelline	C_7_H_8_NO_2_	2.0	138.0562	138.0557	90.74	−5.27	((M + H)—C_4_H_8_FNO_3_)^+^
Guanine	C_5_H_5_N_5_O	2.3	151.0498	152.0571	90.77	−2.79	((M + H)—C_5_H_5_N_5_O)^+^
4-Hydroxycoumarin	C_9_H_6_O_3_	4.6	162.0320	163.0393	83.15	−1.79	((M + H)—C_9_H_6_O_3_)^+^
Dl-tetrahydropalmatine	C_21_H_25_NO_4_	6.5	355.1792	356.1866	88.32	−2.40	((M + H)—C_21_H_25_NO_4_)^+^
Marmesin	C_14_H_14_O_4_	9.8	246.0899	247.0972	80.70	−2.71	((M + H)—C_14_H_14_O_4_)^+^
Phytosphingosine	C_18_H_39_NO_3_	11.4	317.2931	318.3004	87.58	−0.43	((M + H)—C_18_H_39_ NO_3_)^+^
13E-Docosenamide	C_22_H_43_NO	22.1	337.3348	338.3421	97.49	−0.83	((M + H)—C_22_H_43_NO)^+^

Note: Column “Score (lib)” obtained using the MassHunter METLIN software’s Library METLIN database (Personal Compound Database), version 8 and PCDL (Personal Compound Database and Library), version 8.

**Table 2 antioxidants-12-01428-t002:** Proposed chemical components of the Tri-Yannarose recipe’s aqueous extract identified via LC-QTOF-MS in the negative mode based on the match scores obtained during the library search.

Compound Name	Formula	RT (min)	*m*/*z* Calculated	*m*/*z* Observed	Score (Lib)	Mass Error (ppm)	Fragmentation
D-Ornithine	C_5_H_12_N_2_O_2_	1.9	132.0896	131.0823	81.89	2.47	((M-H)—C_5_H_12_N_2_O_2_)^−^
3-Dehydroshikimic acid	C_7_H_8_O_5_	2.0	172.0371	171.0298	93.46	0.49	((M-H)—C_7_H_8_O_5_)^−^
Citric acid	C_6_H_8_O_7_	2.2	192.0266	191.0194	94.45	1.88	((M-H)—C_6_H_8_O_7_)^−^
Succinic acid	C_4_H_6_O_4_	2.7	118.0262	117.0189	99.03	3.77	((M-H)—C_4_H_6_O_4_)^−^
1,2,3-Trihydroxybenzene	C_6_H_6_O_3_	3.1	126.0313	125.0239	91.12	2.83	((M-H)—C_6_H_6_O_3_)^−^
3,4-Dihydroxybenzaldehyde	C_7_H_6_O_3_	3.8	138.0314	137.024	93.26	2.30	((M-H)—C_7_H_6_O_3_)^−^
L-α-Hydroxyisovaleric acid	C_5_H_10_O_3_	4.5	118.0627	117.0555	95.2	2.11	((M-H)—C_5_H_10_O_3_)^−^
Quinic acid	C_7_H_12_O_6_	4.6	192.0636	191.0564	91	−1.26	((M-H)—C_7_H_12_O_6_)^−^
Epicatechin	C_15_H_14_O_6_	5.2	290.0786	289.0713	94.26	1.60	((M-H)—C_15_H_14_O_6_)^−^
Pyrocatechol	C_6_H_6_O_2_	5.6	110.0365	109.0293	89.11	2.29	((M-H)—C_6_H_6_O_2_)^−^
D-(+)−3-Phenyllactic acid	C_9_H_10_O_3_	6.8	166.0624	165.0551	90.27	3.84	((M-H)—C_9_H_10_O_3_)^−^
Methyl N-(a-methylbutyryl) glycine	C_9_H_16_O_4_	7.3	188.1043	187.0971	93.87	2.91	((M-H)—C_9_H_16_O_4_)^−^
Nordihydroguaiaretic acid	C_18_H_22_O_4_	9.8	302.1513	301.1437	82.02	1.71	((M-H)—C_18_H_22_O_4_)^−^
Isopalmitic acid	C_16_H_32_O_2_	21.6	256.2393	255.2321	73.67	3.49	((M-H)—C_16_H_32_O_2_)^−^

Note: Column “Score (lib)” obtained using the MassHunter METLIN software’s Library METLIN database (Personal Compound Database), version 8 and PCDL (Personal Compound Database and Library), version 8.

**Table 3 antioxidants-12-01428-t003:** Total phenolic content, total flavonoid content, and ferric reducing antioxidant power (FRAP) of the Tri-Yannarose recipe and its plant ingredients.

Extracts	Total Contents	
	Phenolic (mg of Gallic Acid Equivalent/g of Extract)	Flavonoids (mg of Catechin Equivalent/g of Extract)
Tri-Yannarose	92.685 ± 0.609 ^b^	14.160 ± 0.472 ^b^
*A. indica*	320.146 ± 7.516 ^a^	76.375 ± 0.204 ^a^
*A. catechu*	92.084 ± 0.129 ^b^	13.206 ± 0.409 ^c^
*T. crispa*	94.360 ± 2.313 ^b^	10.889 ± 0.118 ^d^

^a–d^ Using one-way ANOVA (with Duncan’s test), values in the same column with different superscripts are significantly different in the same medicinal plant (*p* < 0.05).

**Table 4 antioxidants-12-01428-t004:** Free radical scavenging capacities of the Tri-Yannarose recipe and its plant ingredients.

Extracts	Radical Scavenging Properties (IC_50_; µg/mL)	
	DPPH	ABTS
Tri-Yannarose	747.210 ± 44.173 *^,b^	1054.843 ± 151.330 *^,c^
*A. indica*	49.461 ± 3.275 ^a^	197.516 ± 6.036 ^a^
*A. catechu*	373.368 ± 23.082 *^,c^	853.326 ± 160.301 *^,b^
*T. crispa*	1246.322 ± 56.174 *^,d^	2288.524 ± 49.056 *^,d^
Trolox	22.102 ± 8.425 ^a^	158.834 ± 3.246 ^a^

* Significant difference at *p* < 0.05, compared with the positive control. ^a–d^ Using one-way ANOVA (with Duncan’s test), values in the same column with different superscripts are significantly different in the same medicinal plant (*p* < 0.05).

**Table 5 antioxidants-12-01428-t005:** Antioxidant capacities of the Tri-Yannarose recipe and its plant ingredients as determined by single-electron-transfer-mechanism-based assays.

Extracts	Single-Electron-Transfer-Mechanism-Based Assays	
	NBT (IC_50_; µg/mL)	FRAP Assay (mM of FeSO_4_/mg of Extract)
Tri-Yannarose	389.050 ± 3.110 *^,c^	730.541 ± 25.532 ^c^
*A. indica*	108.625 ± 26.323 *^,b^	11,034.986 ± 207.094 ^a^
*A. catechu*	88.921 ± 10.014 *^,b^	933.675 ± 16.307 ^b^
*T. crispa*	396.812 ± 15.204 *^,c^	673.732 ± 12.006 ^c^
Catechin	8.102 ± 1.165 ^a^	NA

* Significant difference at *p* < 0.05, compared with the positive control. ^a–c^ Using one-way ANOVA (with Duncan’s test), values in the same column with different superscripts are significantly different in the same medicinal plant (*p* < 0.05).

**Table 6 antioxidants-12-01428-t006:** Antioxidant capacities of the Tri-Yannarose recipe and its plant ingredients as determined by hydrogen atom transfer (HAT)-based assay.

Extracts	Values of ORAC (µM of TE/µg of Extract)
Tri-Yannarose	0.605 ± 0.266
*A. indica*	0.183 ± 0.918
*A. catechu*	0.844 ± 0.194
*T. crispa*	8.710 ± 1.559

All values are shown as mean ± standard deviation.

**Table 7 antioxidants-12-01428-t007:** The metal chelating activity of the Tri-Yannarose recipe and its plant ingredients.

Extracts	MCA Assay (IC_50_; µg/mL)
Tri-Yannarose	702.109 ± 73.005 *^,b^
*A. indica*	2045.386 ± 28.467 *^,c^
*A. catechu*	2147.625 ± 532.161 ^c^
*T. crispa*	670.110 ± 85.019 *^,b^
Catechin	11.485 ± 0.240 ^a^

* Significant difference at *p* < 0.05, compared with the positive control. ^a–c^ Using one-way ANOVA (with Duncan’s test), values in the same column with different superscripts are significantly different in the same medicinal plant (*p* < 0.05).

## Data Availability

The data used to support the findings of this study are included within the article.

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
