# Peer review of "Evaluation of Chemical Compositions and the Antioxidant and Cytotoxic Properties of the Aqueous Extract of Tri-Yannarose Recipe (Areca catechu, Azadirachta indica, and Tinospora crispa)"

_antioxidants, 2023, doi:10.3390/antiox12071428_

Round 1

Reviewer 1 Report

In this article, a study is made of a traditional Thai herbal medicine formula, but in fact the manuscript prepares the herbal medicine formula, that is, the herbal medicine of any specific brand is not evaluated. In general, the article needs to review the IC50 measurement units (convert all to μg/mL) and refer to what is considered high, medium and inactive for antioxidant activity. Finally, the mass analyzes need to contain more data, so that we can check the identified compounds. Here are some suggestions.

 The authors write “anti-oxidant” in many parts of the manuscript. Correct for “antioxidant”.

 The article is about the study of the Tri-Yannarose recipe and in the “Introduction” of the article reference number 2 is cited. However, this reference is not about the Tri-Yannarose, but about the Tri-Jannarose, which according to this reference 2, has a very similar composition to Tri-Yannarose, but differs in one species, while Tri-Yannarose has Azadirachta indica, Tri-Jannarose has Tinospora cordifolia. But this reference 2 contradicts itself when the study uses Azadirachta indica in the composition of Tri-Jannarose. That is, it is not possible, citing this reference, to know what is right as a composition of Tri-Yannarose. The authors need to cite another credible reference that contains the plants species of Tri-Yannarose.

 The title of the manuscript does not reflect what is in it, because the Tri-Yannarose recipe is made from an aqueous extract of the plants. I suggest rewriting the title.

 Line 59 page 2: I did not understand what the authors meant in this sentence:”However, the medicinal benefits from the trunk of AC have been rarely reported.”

 Line 112 page 2: In the sentence “ According to the Thai medicinal literature”  what is the reference? I didn't understand the following: were the aqueous extracts performed separately and then put together to prepare the Tri-Yannarose? The preparation part of the formula was confusing. I suggest rewriting this part.

 Line  120 page 2: What are the conditions of the QTOF MS?

 Lines 257 and 260 page 6: In the LC-QTOF-MS tables, you need to report the error. Furthermore, why in Table 1 is only the “%Area” for phytosphingosine. In addition, the text preceding table 1 informs that the substances were identified by fragmentation, so tables 1 and 2 must also contain the main fragmentations and the references on which they were based. The supplementary material must contain the mass spectra of each substance identified.

 Lines 284 to 293 page 7: The data reported in the IC50 manuscript are in the mg/mL unit of measurement, and are extremely high values, indicating low antioxidant activity. The IC50 needs to be informed in μg/mL. Furthermore, the paragraph is poorly written as it does not explain when it is describing ABTS results. The same comment follows for the data in Tables 5 and 7.

 Line 368 to 370 page 9:  The manuscript brings decontextualized information in the sentence “However, there was only one study of Tri-Jannarose recipe about the antioxidant activity”. What does Tri-Jannarose have to do with Tri-Yannarose? The reader will be confused. I also emphasize that reference 2 has problems. The article needs to explain somewhere the difference between Tri-Jannarose and Tri-Yannarose.

 Lines 378-380 page 10: Which reference classifies this IC50 content as significant antioxidant activity. What is a significant IC50?

Author Response

The Editor: Antioxidants

June 29, 2023

Dear Sir,

We are submitting the revised version of our manuscript entitled “Evaluation of chemical compositions, anti-oxidant and cytotoxicity properties of aqueous extract of Tri-Yannarose recipe and its plant ingredients” (original title) to be considered for publication in Antioxidants. The manuscript has been revised according to the suggestions and comments of the reviewers. The changes are highlighted in green for comment’s Reviewer1 and yellow for comment’s Reviewer2 in the revised manuscript.  

We hope that the revised version of the manuscript is now acceptable for publication in your journal. I look forward to hearing from you soon.

Sincerely yours,                      

Soiphet Net-anong

Corresponding author: E-mail: soiphet.ne@wu.ac.th 

Reviewer 2 Report

The manuscript “Evaluation of chemical compositions, antioxidant and cytotoxicity properties of aqueous extract of Tri-Yannarose recipe and its plant ingredients” is devoted to phytochemical investigation of Thai Traditional medicine “Tri-Yannarose”.

I do not understand the novelty of this study. All three plants (Areca catechu, Azadirachta indica, Tinospora crispa) are well studied, the antioxidant activity of all three has been investigated. The authors did not provide any new data on these species. In addition, it is not clear how the components were identified by the LC-QTOF-MS method. All of the compounds shown in Tables 1 and 2 are general and nor-targeted for these plant species. The chromatogram of Tri-Yannarose is not shown. There is also a lot of unreliable data.

Other comments:

1.      Lines 47–48. “Antipyretic, diuretic, expectorant, appetizing, and nourishing elemental fire” are not diseases.

2.       Where is reference [3]?

3.      Line 66. “This phytochemistry plant”. What do the authors mean?

4.      Lines 85–88. The authors cannot compare their data on Areca catechu inner trunk with those of other investigators on Areca catechu seeds. If the authors studied both the inner trunk and the seeds, then a comparative analysis could be carried out.

5.      Line 112. Why were fresh materials coarsely ground? Why is the particle size not listed?

6.      It is not clear which solution is involved. There is no sample preparation for research.  Sample preparation for each study should be described.

7.      MS conditions are not fully described.

8.      Lines 134, 136. 120 L, 125 L. May be µL?

9.      Lines 165, 175, 195. The same.

10.   Table 1 and 2. It is not clear how the identification of compounds was carried out. Why is there no chromatogram? The data is unreliable. Please provide a chromatogram.

11.  Table 3. How can Tri-Yannarose IC50 be 92? Tri-Yannarose has three components in composition, the IC50 of each is above 92. The IC50 must be within 167. A similar situation is with the content of flavonoids. The data are unreliable.

12.  Table 5. Why do the authors duplicate data on FRAP-analysis from Table 3?

13.  Table 7. The same situation with calculations as in Table 3.

14.  Lines 358–360. The authors cannot discuss the antioxidant activity of the components in Tri-Yannarose, since their identification was not carried out properly.

15.  Lines 372–374. What early studies were discussed by the authors?

16.  Line 376. “both the betel nut and the betel nut”.

17.  Lines 426–429.  Organic acids do not have pronounced antioxidant properties. 

Author Response

(The authors gave the same response as above.)

Round 2

Reviewer 1 Report

 The authors have made several modifications, but there is still some information to be clarified.

Introduction:

It is still not clear to me the citation of reference 2. Any reader who consults reference 2 will not be able to correlate with this study, since reference 2 mentions the name Tri-Jannarose, while in the manuscript it brings the study of Tri-Yannarose. This explanation (lines 401 to 405) should be in the introduction. So, in line 49 “A previous study discovered TYN (Areca catechu (seed), Azadirachta indica (root), and Tinospora crispa (vines))” this reference is not related to TYN (Tri-Yannarose) but to Tri-Jannarose.

Tabel 1 and 2:

Since Table 1 shows the data in positive mode, the column "formula" must be of the positive ion, the column "molecular weigth" must be replaced by the “m/z calculated”, the column “m/z” must contain the name “m/z observed”. There is still a column missing informing the main fragments. Placing the spectra in the supplementary material does not exempt you from placing the main fragments in this table. In the  column "Score (lib)" put a note in the table about which library was used. The “RT (min)” column must be first, and must be in ascending order retention time. In addition, for RT use only one decimal place after the decimal point. In this table, the compound “Trimeprazine” presented a high error and a low score. What made you conclude that it is compound?

The table title needs to be modified. The word "discover" is a strange term. Replace with "Proposed identification of  Tri-Yannarose recipe's aqueous extract by LC-QTOF-MS in a positive mode based on match score for the library search "

The suggestions for table 1 are the same as those for table 2.

Line 134: The collision energy is 10, 20 and 40 eV not 10, 20 and 40 eV. If there was fragmentation, Tables 1 and 2 should contain the fragmentation column and the supplementary material should contain the fragmentation spectra. The authors included in the supplemental material only the high-resolution mass spectra. You also need to put the fragmentation ones. In addition, some chemical bonds of structures drawn in the supplemental material are odd.

 Line 226: Check if this was the unit of measurement as article 31 uses μM. Check the displayed units of the results (μg/mL or g/mL).

Line 418-420: I don't agree with that sentence “This study is similar to a previous study reported that DPPH activity study of the aqueous extraction of Tri-Yannarose recipe showed 1,485 ± 5 μg/mL”. I'd rather replace Tri-Yannarose recipe with Tri-Jannarose.

Lines 420 to 426:  Based on what evidence did you conclude what is written in these sentences? Because, when compared with the control, the results are very distant. Look for an article that classifies antioxidant activity based on IC50 value for extracts.

Line 428 to 431: I don't agree with that sentence “The other study of the aqueous extraction of aqueous extract of Tri-Yannarose recipe presented lower total phenolic contents at 51.15 ± 0.421 mgGE/gExt and total flavonoid contents showed 0.0002 ± 2.857 mgQE/gExt [2]”. I'd rather replace Tri-Yannarose recipe with Tri-Jannarose.

Author Response

The Editor of Antioxidants

July 6, 2023

Dear Sir,

We are submitting the revised version of our manuscript entitled “Evaluation of chemical compositions, anti-oxidant and cytotoxicity properties of aqueous extract of Tri-Yannarose recipe and its plant ingredients” (original title before revised) to be considered for publication in Antioxidants. The manuscript has been revised according to the suggestions and comments of the reviewers. The changes are highlighted in green for comment’s Reviewer1 and yellow for comment’s Reviewer2 in the revised manuscript. Please see the attachment. 

We hope that the revised version of the manuscript is now acceptable for publication in your journal. I look forward to hearing from you soon.

Sincerely yours,                              

Soiphet Net-anong

Corresponding author: E-mail: soiphet.ne@wu.ac.th 

Reviewer 2 Report

- The novelty of the study remains unclear. Add relevant data in the Introduction.

- The authors did not indicate the particle size of the raw material.

- Table 4. In an earlier version of the article, the value was 0.147 (147).

- Recalculate the values of SD. The data are unreliable with your values.

Author Response

(The authors gave the same response as above.)
